## Classics

dryland vegetation patterns; epidermal tissues; pattern formation; phyllotaxis; Rho-of-Plants.

**Corresponding author:**
Eva E. Deinum
Email: eva.deinum@wur.nl

**Associate Editor:** Ari Pekka Mahönen

# The Turing heritage for plant biology: all spots and stripes?

Eric Siero[1] and Eva E. Deinum[1] 

[1]Mathematical and Statistical Methods (Biometris), Wageningen University & Research, Wageningen, 6708PB, The Netherlands

## Abstract

In 'The chemical basis of morphogenesis' (1952), Alan Turing introduced an idea that revolutionised our thinking about pattern formation. He proposed that diffusion could lead to the spontaneous formation of regular patterns. Here, we discuss the impact of Turing's idea on plant science using three well-established examples at different scales: ROP patterning inside single cells, epidermal patterning across several cells and whole vegetation patterns. Also at intermediate levels, e.g., organ spacing, plants look surprisingly regular. But not all regular patterns are Turing patterns, careful observation and prediction of the patterning process—not just the final pattern—is critical to distinguish between mechanisms.

## 1. Introduction

In 1952, Alan Turing put forward a model that, without diffusion, remains in a stable spatially homogeneous steady state, but with diffusion can spontaneously form regular patterns. We now call this a reaction-diffusion system. The idea was revolutionary, as diffusion was previously understood as a great equalizing process, that blurs and erases, not *creates* patterns. With 'The chemical basis of morphogenesis' (Turing, 1952), Turing was a front runner in applied mathematics (Dawes, 2016), lead the way to development of the concept of dissipative structure (Prigogine & Nicolis, 1967, see glossary), helped explain oscillatory patterns in the Belousov–Zhabotinsky reaction (Zaikin & Zhabotinsky, 1970) up to the eventual observation of stationary Turing patterns in a chemical reaction (Castets et al., 1990) and spurred activity in mathematical and developmental biology (e.g., (Murray, 2003)).

As can be seen from the citation graph (Figure 1), however, there was a significant delay before the idea of Turing was picked up and became mainstream. Two events did not help with the promotion. First, the discovery of the double-helix structure of DNA in 1953 (Franklin & Gosling, 1953; Watson & Crick, 1953; Wilkins et al., 1953) lead to a focus on genetics rather than self-organisation (Ball, 2015). Second, with Turing's death in 1954, the main proponent left the scene. Turing himself envisioned application to plant science, which is the focus of the present article. Much of the early advancement was left in the hands of the botanist C.W. Wardlaw, with whom Turing discussed applications of his theory to plant science, including phyllotaxis (e.g., Wardlaw, 1953).

Some seventy years down the line, we have learned that phyllotaxis is far more complicated than the simplest reaction-diffusion system exhibiting Turing patterns, with modern models including directed transport of the plant hormone auxin via dynamically positioned, polarly localised PIN proteins and often more (Hartmann et al., 2019; Jonsson et al., 2006; Reinhardt & Gola, 2022; Smith et al., 2006). In simple contexts with static PIN distributions, auxin movement can be mapped onto a diffusion-advection model (Boot et al., 2016; Mitchison, 1980). At a more abstract level, moreover, the PIN/auxin module (the 'up the gradient' phenomenological rule for PIN polarisation as used in (Jonsson et al., 2006; Smith et al., 2006) and others) has been compared to a Turing system and can produce all standard types of Turing patterns by varying relevant parameters. In this, the polarisation of PIN proteins towards existing primordia produces inhibitory fields of low auxin around them, thus fulfilling the role of 'long range inhibitor' (Sahlin et al., 2009). So, if phyllotaxis is governed by a Turing instability at all, this certainly requires a liberal definition of it. Once the idea was out there, however, plant scientists started seeing spots and stripes in many different areas of plant science.

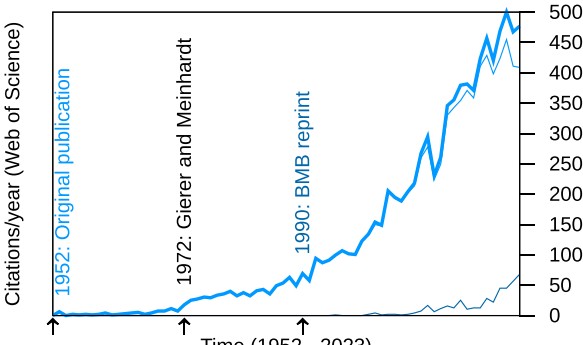

**Figure 1.** Annual citations to Turing's paper ((Turing, 1952) and (Turing, 1990) combined) according to Web of Science. Citations to individual editions in thin lines, with shades of blue as indicated in the graph. Thick line shows the sum. Data retrieved on 23 August 2024.

Here, we address the influence of (Turing, 1952) on plant biology with help of three prominent examples of likely Turing systems and a counterexample. But first we revisit the popular interpretation of Gierer and Meinhardt (Gierer & Meinhardt, 1972), whom at the time of submission were unaware of Turing's publication (Meinhardt, 2012). For a basic mathematical analysis of Turing instability we refer to Segel and Jackson (1972) that appeared in the same year as (Gierer & Meinhardt, 1972).

## 2. The basic concept

Gierer and Meinhardt put forward the intuitive concept of short range facilitation and long range inhibition. In the simplest case, only two components are needed, with different diffusion coefficients (Figure 2a and b).

The slowly diffusing component is self-activating, meaning that perturbations from equilibrium are reinforced by the component itself. The self-activation can result from auto-catalysis or facilitation (see glossary). This component is often referred to as the activator and because of the slow diffusion, facilitation is only short range.

The other component has a larger diffusion coefficient and is, in a mathematical sense, self-inhibiting. This means that perturbations from equilibrium are reduced by the component itself. If this component also inhibits the activator, it is referred to as an inhibitor, if it is required to produce the activator, it is referred to as a depleted substrate. Mathematically, there is no significant distinction between inhibitor and depleted substrate, because one can easily transform one form into the other (Figure 2a and b: Flipping the shallow curve around the x-axis interconverts the graphs of a and b). For example, in the context of dryland vegetation patterns, water can be regarded as a depleted substrate. If one would replace water by lack-of-water in the model, lack-of-water acts as an inhibitor (Siero, 2020).

The critical wavelength ($l_c$) against which the spatially homogeneous steady state first becomes unstable depends on the diffusion coefficients, but scaling slowly: if both diffusion coefficients simultaneously increase two orders of magnitude the critical wavelength only grows one order of magnitude, or more general $l_c \sim \sqrt{D_1 \cdot D_2 / (c_1 D_1 + c_2 D_2)}$ (Segel & Jackson, 1972, equation (13) with $l_c \sim 1/$wave number $k$). Also parameters without a length dimension can impact the pattern wavelength (e.g., Deinum & Jacobs, 2024). Interestingly, if the spatially homogeneous equilibrium is almost

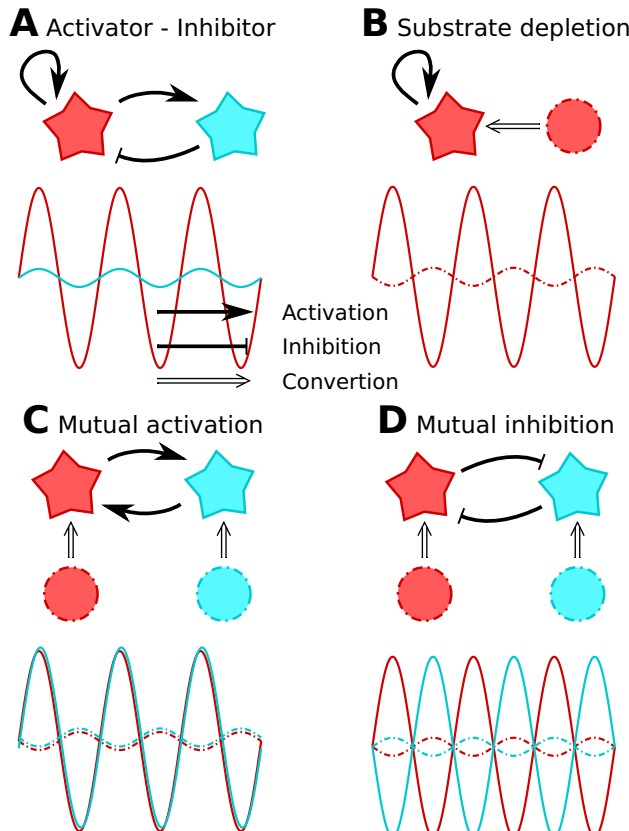

**Figure 2.** Different forms of the Turing system illustrated conceptually. Stars indicate different (active) substances, circles indicate substrates (e.g., inactive forms) of the same coloured substances. Graphs sketch the 1D concentration profiles in steady state. Line styles and colours are the same as the borders of the corresponding substances. Note that the inhibitor/depleted substrate profiles are shallower, due to their relatively higher diffusion coefficients. Cyan graphs for the mutual activation scenario (c) are shifted a little for visualization purposes.

unstable against spatially homogeneous perturbations, the wavelength of emerging patterns can become very large, even with low diffusion coefficients. Note, however, that, contrary to systems with a single localised source, there is no need for individual molecules to travel from one side to the other.

In a given system, it may be difficult to identify components with sufficiently different diffusion coefficients. Here are some known ways to fulfil or circumvent this condition:

- Incorporation of a component with zero diffusion coefficient increases the spread of diffusion coefficients. Complex formation with a non-mobile component can result in an activator with reduced *effective* diffusion (Lengyel & Epstein, 1992).
- Advection is directed and, therefore, more effective at transporting information over longer distances. This could replace or enhance the diffusion of inhibitor/depleted substrate (Rovinsky & Menzinger, 1992). An example of this is formed by dryland vegetation patterns (Klausmeier (1999), Example 3 below).
- If local advection or other processes have no net direction at the length and time scale of the pattern, they can be modelled as relatively fast *effective* diffusion (Pavliotis & Stuart, 2005).
- Mechanical stresses can propagate a signal for more than a few cells (Murray et al., 1988; Heisler et al., 2010).

In three (or more) component systems, Turing instability is possible without a single activator component, e.g. there can be two components that are both not self-activating with a positive feedback loop between them (e.g., Figure 2c and d). There are again two possibilities: they either activate (c) each other or they inhibit (d) each other (Meinhardt, 2012). Mathematically, one can again transform one form into the other. In two-component systems, stationary patterns emerge; if the number of components is larger than two, also oscillatory patterns can emerge (White & Gilligan, 1998). Something else that is possible with more than two components is the appearance of Turing instability in a system with non-mobile components and diffusing components, in which the diffusing components all have the same diffusion coefficient, but interaction with the non-mobile components gives them different *effective* diffusion coefficients as in Lengyel and Epstein (1992).

## 3. Example 1: ROP proteins (within single cells)

Inside single plant cells, we find the evolutionary conserved patterning system of the Rho-of-Plants (ROP) proteins. These proteins are involved in many intracellular membrane patterning processes, specifying a single domain, e.g., in cell polarity and localised (tip) growth, or multiple, e.g., developing puzzle shaped cells (Figure 3a), or complex secondary cell wall patterns in xylem (Deinum & Jacobs, 2024; Müller, 2023; Pan et al., 2023). In the popular (simpli-

fied) description, ROPs have an active, membrane bound state and an inactive, cytosolic state. As diffusion in the cytosol is faster than in the membrane, the requirement for different diffusivities is met and the system can form Turing patterns via a 'substrate-depletion' mechanism (Jacobs et al., 2019).

An important topic in the mathematical literature about ROPs and related small GTPases from fungi and animals is how and when multiple clusters of active ROP can stably, or at least transiently, coexist (Champneys et al., 2021; Deinum & Jacobs, 2024; Goryachev & Leda, 2020; Jacobs et al., 2019). This question is very important for plant biology, as several ROP-controlled patterns strictly require stable coexistence of multiple clusters (Jacobs et al., 2019), such as the numerous lobes formed on leaf epidermal 'pavement' cells (Fu et al., 2002; Fu et al., 2005; Fu et al., 2009), the number of which can even increase during development as the cells grow (Sánchez-Corrales et al., 2018), and the regularly spaced secondary cell wall reinforcements in different xylem types (Higa et al., 2024; Nagashima et al., 2018; Oda & Fukuda, 2012, 2013). Whether a ROP-like model ultimately produces always a single cluster, or allows for the stable coexistence of multiple clusters given a sufficiently large domain, can be understood via cluster level bookkeeping, accounting the amount of active ROP with a single ordinary differential equation per cluster (Jacobs et al., 2019). Both types of patterns can form, however, from an unstable homogeneous steady state through a Turing instability and initially can look very similar.

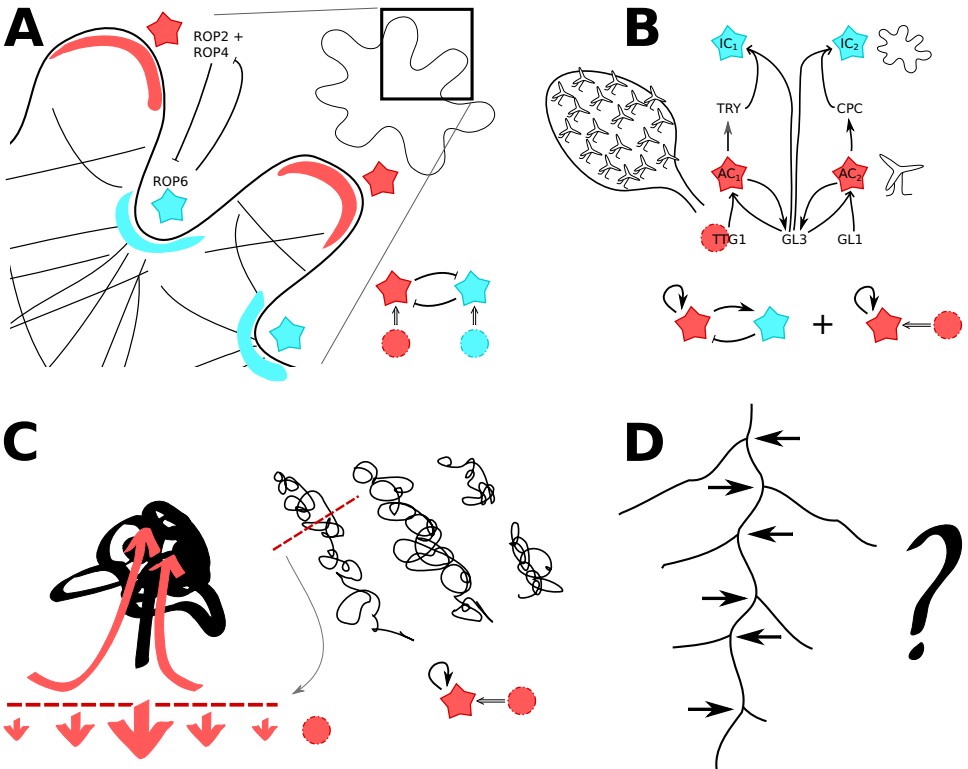

**Figure 3.** Examples of likely Turing patterns in plant biology (a–c). Cartoons with stars and circles refer to Figure 2. (a) ROP proteins govern many cases of local cell growth and other local cell wall modifications, including the puzzle shape of many leaf epidermal cells (Deinum & Jacobs, 2024). In this particular example, multiple ROPs occur in a mutually inhibiting interaction. Each ROP functions via a depleted substrate mechanism, in which the active form diffuses slower than the inactive form (Deinum & Jacobs, 2024). (b) Patterning of trichoblasts (that will produce root hairs or trichomes) and atrichoblast (that will not) in epidermal tissues is governed by multiple transcription factors, here a simplified network for trichomes is shown (Balkunde et al., 2020). IC: inhibitor-complex; AC: activator-complex; other acronyms refer to individual transcription factors. The ones functioning as 'inhibitor' (TRY and CPC) are smaller and can move via plasmodesmata to neighbouring cells (Grebe, 2012). Both activator-inhibitor and substrate depletion models exist for the system. For best explanation of the patterning phenotype of the *ttg1-9* mutant, a combination of activator-inhibitor and substrate (TTG1)-depletion elements is required (Balkunde et al., 2020). In drylands, vegetation and bare soil occur in patterns like spots or stripes (Deblauwe et al., 2008) mediated via local water availability. Water preferentially infiltrates near vegetation, resulting in a net flow from bare to vegetated areas (Ludwig et al., 2005) along the depicted transect. (d) Counterexample: Although lateral roots are regularly spaced along the main root (arrows), it is unlikely that their spacing is controlled by a Turing-like mechanism (van den Berg et al., 2021).

This illustrates the importance of the non-linearities and the far-from-equilibrium dynamics for ultimate model behaviour.

## 4. Example 2: Epidermal tissues (several cells)

Moving up a level, striking examples that are associated with Turing patterns can be found in epidermal tissues. Leaves and other above-ground parts are often covered with stomata for gas exchange and trichomes (Figure 3b) for protection and/or secretion of specialized metabolites. Similarly, root hairs may be restricted to particular cell files in certain species. For optimal function, these cells should appear evenly distributed over the epidermal surface, but where exactly they are formed is of secondary importance. These requirements can easily be solved by a Turing type patterning mechanism, and the underlying patterning mechanisms indeed seem to function as such (Balkunde et al., 2020; Bouyer et al., 2008; Grebe, 2012; Robinson & Roeder, 2015). Root hairs and trichome patterning share many components. Interestingly, some act cell non-autonomously, that is, they move intercellularly via plasmodesmata, whereas others don't. This gives the required basis for a difference in diffusion coefficients. Just as playing with the diffusion coefficients in a plain Turing model changes the wavelength of the resulting pattern, defects in the regulation of plasmodesmatal aperture, which directly affect the intercellular diffusion of 'small' molecules, have severe impact on the patterning of root hairs (Kim et al., 2002) and stomata (Guseman et al., 2010; Kong et al., 2012; Okawa et al., 2023).

## 5. Example 3: Dryland vegetation patterns (multiple plants; 10–100 m)

Contemporaneous with Turing's paper, vegetation patterns (Figure 3c) were reported for the first time (Macfadyen, 1950), but it would take about fifty years before they would be firmly linked. Using satellite imagery, it became apparent that spatially periodic vegetation interspersed with bare soil—creating a spot, stripe, labyrinthine or gap pattern—is present in drylands around the globe (Deblauwe et al., 2008).

Vegetation spreads slowly and increases soil permeability. The water flows relatively fast, until it infiltrates into the soil somewhere (or has evaporated). Because of the difference in permeability, there is a net flow of water from bare ground to vegetated patches (Ludwig et al., 2005). Thus, vegetation locally increases water availability (facilitation) at the cost of water availability in surrounding bare soil (competition).

The first multi-component model explaining dryland vegetation patterns this way did not incorporate water diffusion, but instead downhill flow of water by advection (Klausmeier, 1999). Later models included water diffusion, allowing for Turing instability, and extended to three components by distinguishing between surface and soil water (HilleRisLambers et al., 2001; Gilad et al., 2004). Having advection, spatial pattern formation is possible without different diffusion coefficients. There are fundamental differences though, as advection has a direction and, therefore, introduces anisotropy, which is visible in the selection of stripe patterns and their orientation (Siero et al., 2015). In wetter ecosystems, other factors like nutrient distribution could cause vegetation patterning in similar ways (Rietkerk & van de Koppel, 2008).

## 6. Counterexample: lateral root spacing

Also at intermediate levels, plants look strikingly regular, e.g., in the spacing of lateral roots (Figure 3d), vascular bundles or branches.

Contrary to the canonical Turing pattern, lateral roots and side branches are not formed all at once, but develop one by one as the root or shoot grows. As of yet, no convincing model has been established that matches our other examples in simplicity. One option is that at these levels, there are so many factors affecting the pattern, that the Turing mechanism is hard to discern. Another option is that these processes are regulated in a very different manner that happens to produce regular patterns as well. A thoroughly investigated example of the latter case can be found in the work on lateral root priming by van den Berg et al. (2021). The authors derive from their models a new type of spacing mechanism, dubbed 'reflux-and-growth', and compare this with a 'Turing mechanism' on a unipolarly growing domain and a 'clock-and-wavefront' mechanism (known from the regular sequential specification of somites in vertebrates (Cooke & Zeeman, 1976)). Of these, reflux-and-growth best describes the experimentally observed effects of growth rate on the spacing of primed sites and the timing of their their occurrence. This example demonstrates that there are multiple ways of obtaining a regular pattern, and distinguishing between different mechanisms requires looking beyond single static patterns, e.g., by also investigating temporal dynamics and trends in responses to parameter changes.

## 7. Discussion

With 'The chemical basis of morphogenesis', Turing has brought us a revolutionary new way of thinking about regular patterns in nature. This has tremendously advanced our understanding of plant biology across scales. Nonetheless, it took a long time for his ideas to become widely accepted. Perhaps not surprisingly, because even for something as 'simple' as plain chemical reactions, it took several decades to experimentally observe stationary Turing patterns (Castets et al., 1990).

In much of his manuscript, Turing works with a single ring of 20 cells. He comments that even when *a reasonably complete mathematical analysis was possible, the computational treatment of a particular case was most illuminating.* That is, he immediately recognizes the illustrative power of computer simulations. For non-linear systems the use of computer simulations avoids simplifying assumptions and the disadvantage of only obtaining particular results *'is probably of comparatively little importance'.* This attitude prevails in current day studies of development and other spatial processes.

The use of computer simulations has the added advantage that it is much easier to obtain temporal solutions. The example of lateral root spacing clearly illustrates why this is important: multiple mechanisms exist that can give rise to regular patterns that look like Turing patterns in steady state, but may show very different dynamic behaviour. Different mechanisms may also respond differently to perturbations (including parameter changes). Rigorously testing model predictions on these aspects is particularly important if the actual components fulfilling the roles of activator and inhibitor/depleted substrate are not (yet) known.

At the end of his manuscript, Turing humbly admits that the examples he uses are much simpler than most biological systems, ending with: *'It is thought, however, that the imaginary biological systems which have been treated, and the principles which have been discussed, should be of some help in interpreting real biological forms'.* The complexity of many biological systems does not automatically rule out a Turing mechanism. As described above (Figure 2c and d), the role of a single self-activating component

could actually be fulfilled by two sets of mutually activating or inhibiting components, and/or be facilitated by accessory proteins. Similarly, a single arrow could consist of a long chain of reactions. Upon more detailed observation, both types of complications are found in the ROP system and the epidermal patterning systems, yet, the necessary simplifications for extracting the Turing mechanism are straightforward and easily accepted. Vegetation patterns occur at such high level, that we cannot even imagine that there would be no additional factors involved, yet we very easily accept this level of simplification. Phyllotaxis and other patterns of organ spacing, however, may be in a 'hard' length range for straightforward reaction-diffusion systems. These patterns would be subject to the same biological constraints as the epidermal patterns in Example 2, but it is impossible to scale up the pattern wavelengths by changing the key proteins involved for faster diffusing ones. A ten-fold increase would be required for both 'activator' and 'inhibitor', which, for spherical proteins obeying the Stokes-Einstein relation, would require a 1000-fold reduction in the number of amino acid (AA) residues, whereas the proteins involved are much shorter (Caprice (CPC): 94 AA; Werewolf (MYB66): 203 AA). As a further complicating factor, cell walls act as barriers that reduce the range of effective diffusion coefficients (Crick, 1970; Deinum, 2013). For organ spacing, the cellular nature of the tissue cannot be ignored—it may even be critical to the process replacing diffusion (e.g., Sahlin et al., 2009). This cellularity, however, invites the formulation of rather complex models, including diverse cellular responses, which in turn are very hard to simplify to the extent that the equations reveal the mathematical nature of the patterning mechanism. Although tools exist that help to find Turing instability in larger reaction diffusion systems (e.g. Marcon et al. (2016)), they do not overcome the increased uncertainty about parameter values and model structure inherent to larger models.

All in all, we now know just about enough about phyllotaxis to safely state that Turing picked a very hard problem to start his career in plant science.

## 8. Conclusion

Regardless of whether the mechanism underlying the botanical phenomenon of our interest turns out to be truly caused by a Turing instability, or not, Turing has inspired us to ask questions about what can cause spontaneous patterning behaviour. On the one hand, the great success of the idea does not mean that every regular pattern is a Turing pattern. On the other hand, Turing instabilites are not at all restricted to simple systems. In the details of the pattern specification, as well as the development of the actual objects that follow the pattern, biology shows its full complexity, or, as Turing allegedly has said about the zebra: *'Well, the stripes are easy. But what about the horse part?'*

**Author contributions.** E.S. and E.E.D. together wrote the manuscript.

**Financial support.** This research received no specific grant from any funding agency, commercial or not-for-profit sectors.

**Competing interest.** None

**Data Availability Statement.** Not applicable

**Glossary of terms.**

**Reaction-diffusion system** A mathematical model consisting of multiple components where the time evolution of each component is prescribed by diffusion (spatial spread) and reaction (changes in quantity through interaction with other components).

**Advection** Directed movement of a substance in a certain fixed direction, as opposed to diffusion that results in spreading out equally in all directions.

**Isotropy** Uniformity in all orientations. Diffusion typically is isotropic, advection is not.

**Non-linearity** Term that is non-linear, meaning that doubling of substance quantities does not (always) result in doubling of the relevant rate.

**Auto-catalysis** A (chemical) reaction where the reaction rate increases with the concentration of a reaction product.

**Facilitation** A feedback where presence of a substance stimulates an increase of the substance itself.

**Dissipative structure** In open systems, i.e., where energy or mass can be exchanged with the surroundings, the second law of thermodynamics does not apply: instead of only a spatially homogeneous equilibrium, spatial patterns may persist. These patterns were referred to as dissipative structure by Ilya Prigogine (nobel prize in chemistry in 1977) as they require a continuous exchange of matter and/or energy.

**Far-from-equilibrium** When the state is not close to a spatially homogeneous equilibrium, which in a thermodynamic setting would be a state with minimal energy.

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
