## [Reviewer Report]

In this manuscript, Siero et al has described the impact of the classic paper of Alan Turing on plant research. Recently some reviews has been published highlighting the impact of this paper on research but more focussing on the mathematical side of it. Therefore, this plant focus of this review is refreshing and is an interesting read for everyone in the plant research community. The examples used to illustrate different Turing pattern observed in plants are well chosen and I appreciate that they represent different levels of magnitude (cellular, tissue and field level). The inclusion of the counter example, is a well needed caveat that not every pattern is and should be resulted from a Turing pattern.

I have only a few comments that I envision to be beneficial for broaden the appeal of this paper to an bigger audience.

Major comments:

- I miss an overview figure that illustrates the examples used in the review and their mechanism. If figure number is limited, please then combine Figure 1 and 2 into one.

Minor comments:

- To extend to interest of this paper to non-mathematical researchers. I would suggest to add a glossary of used terms that might not be common knowledge. For example, terms like advection could use a definition.

- Bottom two panels of Figure 2 has been referenced twice in two adjacent sentences, please remove of of these. Additional, I would suggest to add panel numbering to figure 2. For example Figure 2A, to make it easier to direct the readers to the correct panel.

- In the first example of Turing patterns with the ROP proteins. I would prefer the authors to start with describing the Turing patterns created by these proteins (Cell wall and lobes in epidermis) before explaining the mechanism that creates the Turing components. This would endorse people to think more about their own research topic in terms of Turing patterns, which should be the goal of this review.

---

## [Reviewer Report]

The manuscript by Siero and Denium on the Turing heritage for plant biology nicely and concisely reviews some aspects of the Turing patterning and its relation to patterning in plants.

Please find below some minor points to be considered, which I hope the authors can successfully address without too much extra work:

List of minor points:

-Several claims do not have a supporting reference, which it is very important in a review. I would ask the authors to carefully check this point. For instance,

* Historical interpretations of Turing success in page 1 – what historical analyses support the statement referring to the two events not helping with Turing’s ideas?

* Page 2: Statements regarding the insensitivity of the critical wavelength with respect to diffusion coefficients, and when referring to l_c approx. (D_1 D_2)^1/4

* Page 2, the two bullet points referring to advection.

* Page 2, last sentence before example 1, referring to the existence of stationary and oscillatory patterns

*Page 2, when introducing the ROP proteins

-Some clarity could be improved, for instance:

*Statement in page 2 “The self-activation can result from a constant per capita loss rate and a per capita growth rate that increases with size, referred to as facilitation “; the per capita also appears confusing to me, the readership is thinking in terms of concentrations, so does it refer to ‘per molecule’? what does it mean ‘with size’ here?

*Statement in page 2 “Seemingly, this is at odds with the perception that with low diffusion coefficients, the diffusion length is small, but in isotropic systems, there is no need for individual molecules to travel from one side to the other. “

*In example 1 (pages 2 and 3); ‘multiple clusters’ would be equivalent to ‘different concentration maxima’, is that right? Also, ‘pattern coexistence’ refers to ‘cluster coexistence’, i.e. having multiple maxima? Could you briefly explain what ‘cluster level bookkeeping’ mean and involve?

*Sentence in page 2 “Mathematically, there is no significant distinction between inhibitor and depleted substrate, because one can easily transform one form into the other”. Although I see the analogy from the authors, I am afraid that saying ‘Mathematically’ (although you say ‘no significant distinction’) might be going too far and perhaps might lead to confusion, given that this analogy, in my understanding, this is not strictly ‘Mathematically’ true – but I might be wrong. Please revise this, and consider to take out the word ‘Mathematically’.

*Sentence in page 3 “This gives the required basis for a difference in diffusion coefficients”. I guess this refers to the fact of non-mobile factors relaxes the need of having 2 diffusible species with different diffusions, right? Consider to rephrase and be more explicit, and I would suggest cite Marcon et al 2016 eLife.

*Sentence in page 3 “Having advection, spatial pattern formation is possible without different diffusion coefficients.” – but strictly speaking, the different diffusion coefficients are just necessary for two species, is it right? (see Marcon et al 2016 eLife)

*Page 3, in example 2; consider to rephrase the sentence ‘This gives the required basis for a difference in diffusion coefficients.’, given this would be just a requirement for 2-component systems, while some of these models are not 2-component systems anymore

* Sentence in page 4, “ is impossible to scale up these patterns by changing the key proteins involved for faster diffusing ones.” – is scaling up referring to increasing the pattern wavelength?

-Sentence in page 2 “Other factors can also have a big impact on the pattern wavelength. “, I would take out ‘also’, given the previous sentence refers to a factor being quite insensitive to the wavelength.

- Given the advection terms are often mentioned during the text, I would suggest the authors to consider to mention at some point whether the PIN-mediated auxin transport could be also understood as an advective term.

-In Figure 2, there is a typo: “Substate” should be “Substrate”

-Page 3 instead of “Turings paper” should be “Turing’s paper”

---

## [Editor Report]

Dear Dr Siero and Dr Deinum, 

thank you for submitting such an interesting Classics on Turing patterns. I learned a lot while reading it. The two reviewers have a few points mainly to increase the readability of the text.

best wishes,

Ari Pekka

---

## [Reviewer Report]

I would like to thank the authors for the changes they provided to the manuscript to address my comments. I feel like that the manuscript has been greatly improved. Therefore, I have no additional comments and would support the acceptance of the manuscript.

---

## [Reviewer Report]

I would like to thank the authors for the different improvements. Although the manuscript reads better now, I still have a few additional minor suggestions:

- Thanks for the clarification about the rescaling and non-dimensional analysis when referring to l_c approx. (D_1 D_2)^1/4 ; yet, I am wondering whether this relates to a particular or general case or not, and therefore a reference or more clarification would help. E.g., just by dimensional analysis, perhaps one could also argue that l_c approx. (D_2^2/D_1)^1/2, and therefore the conclusions from that paragraph would be different, wouldn’t be?

- In caption of Fig. 2 referring to panel B, it reads like trichomes also come from trichoblasts, which I understand it is not the case. Please rephrase.

- I appreciate the inclusion of Figure 3, which illustrates the different examples discussed within the review. Figure 3 appearing within the text shows that panels A and C are intermingled, which makes panel C difficult to understand. I would suggest to reduce the size of panel A (but this issue is solved when Figure 3 is provided as a separate file and not within the text). I like panel C, but at the same time it looks very artistic, and might be difficult to interpret for certain readers. I would suggest to adapt it and/or make sure the caption is clearly related to it and facilitates more its understanding.

- Page 2, right before the basic concept subheading. I would rephrase the sentence “For a basic mathematical analysis we refer to (…)” as “ For a basic mathematical analysis on XYZ we refer to (…)”

-Page 2, right after auto-catalysis or facilitation I would add “(see glossary)”.

-Page 2, I still do not understand the meaning of the following sentence: “Note, however, that in isotropic systems, there is no need for individual molecules to travel from one side to the other.”, consider to clarify it.

-Page 2 Consider to rewrite the last sentence before Example 1 into “but interact with the non-mobile components to give different effective diffusion coefficients, as in Lengyel and Epstein (1992).”

-Page 5, the new sentence “Tools exist to help to find…” seems to break the flow of the paragraph and could be misinterpreted, so I suggest to remove it or move it elsewhere. Otherwise, given the previous sentence discusses the cellularity effect, one might think that larger reaction diffusion systems could emulate better the effect of the cellular nature of tissues, which is not necessarily the case.

-In the glossary, it appears a sentence defining Isotropy within the advection definition, which looks awkward – either rephrase the sentence and integrate it in the advection definition, or put it aside as a new definition.

-Note that when citing some panels in the text, a comma should be in-between their labels (i.e. you wrote AB and CD in the text, instead of A, B and C, D, respectively.

-Please revise the references in the reference list at the end of the manuscript (note some of them have twice their website, etc.).

---

## [Editor Report]

Dear Eva,

as you can see, one of the two reviewers still have a few minor issues to handle. Please, address the points you see relevant, and I will proceed forward with the manuscript.

with best wishes,

Ari Pekka

---

## [Editor Report]

Dear Eva,

than you for submitting the revised manuscript. Everything appears to be in order, so I am happy to accept the manuscript.

with best wishes,

Ari Pekka